# Patulin Contamination of Citrus Fruits from Punjab and Northern Pakistan and Estimation of Associated Dietary Intake

**DOI:** 10.3390/ijerph18052270

**Published:** 2021-02-25

**Authors:** Kinza Aslam, Shahzad Zafar Iqbal, Ahmad Faizal Abdull Razis, Sunusi Usman, Nada Basheir Ali

**Affiliations:** 1Department of Applied Chemistry, Government College University Faisalabad, Faisalabad 38000, Pakistan; kinzagcuf@gmail.com; 2Natural Medicines and Products Research Laboratory, Institute of Bioscience, Universiti Putra Malaysia, UPM Serdang, Selangor 43400, Malaysia; usunusi.bch@buk.edu.ng; 3Department of Food Science, Faculty of Food Science and Technology, Universiti Putra Malaysia, UPM Serdang, Selangor 43400, Malaysia; nada44basher@gmail.com

**Keywords:** citrus fruits, patulin, dietary intake, variation in patulin, liquid chromatography

## Abstract

This research aims to assess the natural occurrence of patulin (PAT) in selected citrus fruits from central cities of Punjab and Pakistan’s northern cities. A total of 2970 fruit samples from 12 citrus cultivars were examined using liquid chromatography fitted with a UV detector. The detection limit (LOD) and quantification limit were 0.04 and 0.12 µg/kg, respectively. About 56% of samples of citrus fruits from Punjab’s central cities, Pakistan, were found to be contaminated with PAT, with values ranging from 0.12 to 1150 µg/kg in samples from central Punjab cities. Furthermore, 31.7% of samples of citrus fruits from northern cities of Pakistan were contaminated with PAT, with values ranging from 0.12 to 320 µg/kg. About 22.1% of citrus fruit samples had PAT levels greater than the suggested limits established by the European Union (EU). The dietary intake levels of PAT ranged from 0.10 to 1.11 µg/kg bw/day in the central cities of Punjab, Pakistan, and 0.13 to 1.93 µg/kg bw/day in the northern cities of Pakistan.

## 1. Introduction

Citrus fruits belonging to the *Rutaceae* family, covering 130 genera, are well-known worldwide due to their diversified potential in fruits, juices, confectionaries, and fresh fruit consumption. Citrus fruits can be cultivated in tropical, subtropical, and temperate environmental conditions, covering 137 countries. The essential citrus fruits are oranges, tangerines, limes, mandarins, grapefruits, lemons, citrons, and many hybrid varieties, with pleasant flavors, aromas, and attractive colors. Citrus fruits are enriched with many health-promoting bioactive compounds such as ascorbic acid (vitamin C), phenolic acids, flavonoids, carotenoids, and pectin. Furthermore, calcium, phosphorous, iron, potassium, zinc, copper, and sodium are also present in citrus fruits [1,2]

Pakistan is ranked the 10th country for the production of citrus fruits in the world. The high water content and nutrient composition of citrus fruits make them susceptible to infections by microbial pathogens during harvest, transportation, and storage until consumption. The presence of mold in fruits and juices is a critical issue due to its implication in health care and the economy. Pakistan is known to produce good-quality fruits worldwide. The fruits are grown in tropical and subtropical climatic conditions and are available throughout the year. The fruits are produced in an area of 800.000 hectares, with a production of about 7.05 million tons. Almost 10% of fruits were exported during the 2017–2018 crop seasons [3]. Postharvest diseases in citrus fruits are responsible for massive economic losses throughout the world. It is estimated that 40% of whole citrus fruits produced in Pakistan are wasted during storage or industrial processing. The fungi reduce the shelf life and affect the acceptability of fresh citrus fruits, leading to the rejection of fruits [4,5].

Mold growth in citrus fruits leads to the production of hazardous chemical compounds known as mycotoxins [6]. Mycotoxins are secondary metabolites produced during pre- and postharvest conditions. Mycotoxins are highly diversified, with about 450 different structural categories already identified and classified [7,8,9]. The most important mycotoxins found as food contaminants include aflatoxins, ochratoxin A, fumonisins, zearalenone, trichothecenes, and patulin [10]. Studies have shown that mycotoxins may attain high accumulation levels [7,8] and are resistant to different processing conditions such as fluctuations in pH, water content levels, ions, temperature variations, heating rates, and heating times. The Food and Agriculture Organization (FAO) has documented that almost 25% of food is contaminated with mycotoxins [11]. 

Patulin (PAT, 4-hydroxy-4H-furo [3,2-c] pyran-2(6H)-one) is an unsaturated heterocyclic lactone, known as a toxic class of mycotoxins. It is a common contaminant of fruits such as apples, cherries, maybush, kiwi fruits, strawberries, grapes, mango, pears, apricots, tomatoes, etc. The fungi *P. expansum*, *Penicillium*, *Byssochylamys, Aspergillus*, and *Paecilomyces* are major PAT producers. The highest stability of these toxins is at pH 4.0; therefore, the ripening stage of these fruits is ideal for the attack by PAT-producing fungi. The fungi penetrate the fruits through rapture in the peel, and from there, they spread and contaminate the whole fruit. The accumulation of PAT is accelerated in fruits during storage periods [12,13,14,15]. The high solubility of this toxin makes its transfer easy from fruits to juices during processing. PAT is stable in an acidic medium; therefore, a considerable amount is transferred during the processing of fruits into juices. [16]. Previous studies have documented that PAT induces various health hazards to human health, implicating many acute and chronic disturbances. The acute symptoms include convulsions, dyspnea, agitation, edema, ulceration, pulmonary congestion, GI tract distension, intestinal bleeding, intestinal inflammation, hyperemia, epithelial cell degeneration, kidney damage, vomiting, and other gastrointestinal issues. Furthermore, PAT is immunotoxic, immunosuppressive, neurotoxic, teratogenic, and genotoxic [17]. The International Agency for Research on Cancer (IARC) ranked PAT in Group 3 (not carcinogenic to humans) [18]. Many countries have established maximum permissible amounts of PAT in food products. The Joint FAO/WHO Expert Committee on Food Additives (JECFA) has a short-term maximum acceptable dietary intake (PMTDI) of 0.40 µg/kg body weight (bw)/day [19]. The Codex Alimentarius Commission (2003) [20] has set a maximum limit of 50 µg/kg in apple juice. The European Union (EU) has set the maximum acceptable PAT level at 50 µg/kg for fruit juices, fruit nectars, concentrated fruit juices, cider, alcoholic drinks, and other fermented drinks produced from apples or apple juice. A limit of 25 µg/kg has been implemented in solid apple fruits and 10 µg/kg for apple-derived products and young children and infants [21]. No regulation has been established for PAT in fruits or juices in Pakistan [22].

In a previous study [22], considerable PAT amounts were found in juices, smoothies, and fruits from Pakistan. However, the PAT contamination of citrus fruits in Pakistan requires more extensive investigation because of the importance of this crop in this country. The present survey is designed to analyze the levels of PAT in citrus fruits from Pakistani Punjab and northern Pakistan areas, with reference to EU permissible limits, and to evaluate the daily intake of PAT by local inhabitants. We anticipate that our work will help consumers become aware of the risk and stimulate lawmakers to introduce appropriate regulations.

## 2. Materials and Methods

### 2.1. Samples

A total of 2970 fruit samples from 12 citrus cultivars (kinnow, orange, grapefruit, bitter orange, mausami, red blood, pineapple, sweet orange, rough lime, sweet lime, kagzi lime, and lemon) were collected from major citrus-producing areas of Pakistan (Toba Tek Singh, Sargodha, Multan, Jhang, Sawat, Peshawar, Mirpur) between September 2019 and December 2019. There was a random collection of samples, and each sample’s size was maintained at 1 kg. The samples were stored in plastic bags with proper labeling and kept at −20 °C in a freezer until further analysis.

### 2.2. Reagents and Chemicals

Patulin standard with the concentration of 100 µg/mL in acetonitrile was obtained from Sigma-Aldrich, St. Louis, USA. Ethyl acetate, sodium carbonate, and acetic acid were obtained from Sigma-Aldrich (Lyon, France). The solvents, including methanol and acetonitrile, were of HPLC grade and obtained from Fisher Chemicals (Illkirch-Graffenstaden, France). PAT’s standard curve was made using the methanolic solution at the concentration of 0.4 to 300 μg/L and stored in sealed vials at −20 °C until further analysis. The other reagents and solvents were all of analytical grade.

### 2.3. The Cleanup Procedure

Patulin extraction from the citrus fruit samples was achieved following the method described by Iqbal et al. [22], with some modifications. About 10 g of solid citrus fruit (edible part) sample was added in 10 mL of water and homogenized. Then, 20 mL of ethyl acetate was added and mixed with a vortex mixer for 3 min. After that, the mixture was centrifuged (5 min at 25 °C) at 4500 rpm. The upper organic layer was transferred to a centrifuge tube with the addition of 10 mL of 1.5% (*w*/*v*) sodium carbonate and vigorously mixed. Again the addition of 5 mL of ethyl acetate with vigorous shaking for 5 min was done. The solution pH was maintained at the value of 4.0 with the addition of a few drops of glacial acetic acid followed by evaporation using a nitrogen stream at 60 °C. A 5% solution of acetonitrile (in water) (5 mL) was then added to the residue and filtered with a 0.22 mm syringe filter (Millipore, Burlington, MA, USA), followed by complete drying using nitrogen stream. Finally, the residue was dissolved in 500 µL of methanol, and 20 µL of this solution was injected for HPLC analysis.

### 2.4. HPLC Conditions

Patulin analysis of the citrus fruits was achieved via HPLC (Shimadzu LC-10A series, Kyoto, Japan) equipped with a UV detector (276 nm). A C18 column (set at 25 °C) (4.6 × 250 mm, 5 µm) Discovery (Supelco, Bellefonte, PA, USA) was used for evaluation. The mobile phase was 90% acetonitrile (in water), with a 1 mL/min flow rate in the isocratic mode.

### 2.5. Assessment of Daily Intake

The procedure for estimating dietary consumption in the citrus fruits was followed by our previously established method, i.e., that of Iqbal et al. [22]. The dietary consumption of patulin in the local consumers was calculated from a questionnaire-based survey of local individuals’ eating habits over the former four weeks. Of 500 individuals, 450 returned the questionnaire with answers, 45 individuals did not, and 5 questionnaires were dismissed. The bodyweight of the male individuals was 70 ± 2, and age varied from 32 to 45 years old. The daily intake was assessed utilizing the following procedure
(1)$$\mathrm{Dietary}\;\mathrm{intake}\;µg/\mathrm{kg}/\mathrm{day}=\;\frac{\mathrm{Daily}\;\mathrm{intake}\;\mathrm{of}\;\frac{\mathrm{fruits}}{\;\mathrm{juices}}\;\left(\frac{g}{\mathrm{day}}\right)\times \mathrm{Patulin}\;\mathrm{in}\;\frac{\mathrm{fruits}}{\;\mathrm{juices}}\;\left(\frac{µg}{\mathrm{kg}}\right)\;}{\mathrm{Average}\;\mathrm{individual}\;\mathrm{weight}\;\left(\mathrm{kg}\right)\;}$$

### 2.6. Method Validation

The method was validated through parameters including precision, linearity, reproducibility, limit of quantification (LOQ), limit of detection (LOD), repeatability, and reproducibility. The LOD was obtained from a 3:1 signal-to-noise ratio, and the LOQ was quantified by a 10:1 signal-to-noise ratio [23]. In the recovery analysis, three fortified PAT levels were spiked to uncontaminated orange samples.

### 2.7. Statistical Analysis

PAT data in selected citrus fruits were presented as mean ± standard deviation, and all the samples were analyzed in triplicate. The straight-line equation and coefficient of determination R^2^ were calculated using the simple linear correlation/regression analysis. A Student’s *t*-test was applied to estimate significant differences among different citrus-producing locations using SPSS (IBM Corporation, Armonk, NY, USA).

## 3. Results and Discussions

### 3.1. Validation of Method and Quality Control

Method validation was achieved through quality control parameters. The recovery rate was obtained by spiking blank samples with three patulin concentrations: 50, 100, and 200 µg/L. Mean values of the recovery varied from 85.5 to 91.5%. The average relative standard deviation (RSD) variation ranged from 10.6 to 14.5%, as presented in Table 1. The recovery range was within the requirement (70–110%) recommended by EC regulation 401/2006 [24]. PAT’s standard curve, ranging from 0.4 to 300 µg/L, confirmed the linearity, with a coefficient of determination (R^2^) value of 0.9947. The LOD and LOQ of PAT were 0.04 and 0.12 µg/kg, respectively. These values of LOD and LOQ are lower than the ones reported by [25], which comprised a LOD and LOQ in a sample matrix of 0.5 μg/L and 2 μg/L, respectively. The accuracies of these quantities mainly depend on the sensitivity of the method [26]. The chromatograms in Figure 1a–d show the natural occurrence of PAT in orange, sweet orange, lime, and grapefruit, respectively.

### 3.2. Occurrence of PAT in the Citrus Fruit Samples

The incidence levels of PAT in citrus fruit samples from central cities of Punjab, Pakistan, are reported in Table 2. The results show that 56%, i.e., 1101 out of 1967 samples, were positive for PAT. About 51.4%, 56%, 66%, and 50% from Toba Tek Singh, Sargodha, Multan, and Jang cities were contaminated with PAT, with levels up to 1150 µg/kg in sweet orange samples from Multan city.

The occurrence of PAT in citrus fruits from northern cities of Pakistan is represented in Table 3. The results documented that 372 out of 1003, i.e., 37.1% of samples, were positive with PAT. About 36.7%, 36.8%, and 37.8% of the samples from Mirpur, Peshawar, and Swat were found positive. The highest mean level of 163.3 ± 10.1 µg/kg was found in orange samples from the Swat District of northern Pakistan. About 657 out of 2970, i.e., 22.1% samples were found to exceed the EU permissible limit (Table 4). The PAT levels in citrus fruits from Pakistani Punjab and northern Pakistan were significantly different by applying ANOVA (*p* < 0.05).

The present survey results show that PAT contamination in citrus fruits is much higher than that previously reported for mango and orange fruits [27]. In that case, 74 out of 141 samples of orange fruits, juices, pulp, and orange jams were found to contain PAT, and only in one sample did the level exceed 50 µg/kg. The high PAT contamination of citrus fruits in Pakistan probably reflects that this country does not enforce good agricultural practices, and fungicide is not routinely applied during fruit maturation. During the ripening stage, wounds on the skin of fruits provide preharvest contamination of fruits. Interestingly, PAT levels as high as 113342 µg/kg were detected in apples’ rotten areas [28]. In our previous study [22], 136 out of 237 (57.4%) samples of juices, smoothies, and fruits were contaminated with levels up to 1100 µg/kg. A mean level of 921.1 ± 22.4 µg/kg was found in red globe grapes, and 33.8% of samples were found to have levels higher than the EU recommended limit. Funes and Resnik [29] reported that 21.6% of samples of solid and semisolid apple and pear products were contaminated with PAT, with average levels of 17–221 µg/kg (mean levels of 61.7 µg/kg). The elevated amount of PAT was observed in apple puree (123 µg/kg), and almost 50% of samples were observed to be positive with PAT [29].

Cho et al. [30], from South Korea, analyzed 72 samples of fruits, including three apple, two orange, and four grape samples, and reported a maximum mean level of 30.9 µg/L in orange juice samples. Spadaro et al. [31], from Italy, showed that 47 out of 135 samples of different juices were contaminated with PAT, with a mean level of 6.42 ± 4.48 µg/L and a maximum level of 55.4 µg/L. In another study from Greece, Moukas et al. [32] reported PAT levels in orange juice samples ranging from 3.1 to 10.8 µg/kg.

Zouaoui et al. [33], from Tunisia, analyzed 214 samples (including concentrated juice, apple juice, pear juice, mixed juice, compote, apple jam, and pear jam samples). They documented PAT occurrence in 50% of the analyzed samples, with concentrations ranging from 2 to 889 µg/L and 22% of samples exceeding the EU limit. Murillo-Arbizu et al. [34] found that 66% of apple juice samples from a Spanish market were contaminated with PAT (LOD of 0.7 µg/L), with a mean level of 19.4 µg/L and levels ranging from 0.7 to 118.7 µg/L. They reported that 11% of the samples had PAT levels above the permissible limits of the EU regulation. Similarly, very high levels of PAT were reported by Saxena et al. [35] in Indian-branded juices, including concentrates of apple, orange, guava, grape juice, etc., with values in the range of 21–1839 µg/L and an average level of 330 ± 141 µg/L. The maximum temperature of northern Pakistan remains around 25 °C during summer, and in winter it drops below 0 °C; in contrast, the temperature in Punjab remains in the range of 35–45 °C during summer and fluctuates between 7 and 15 °C in winter, thus accounting for the higher incidence of PAT contamination in this region.

PAT results in different cultivars of citrus fruits are more comprehensive interims of samples analyzed and from central and northern areas were covered during analysis. However, only two reports [22,27] are documented demonstrating the presence of PAT in fruits and juices. In our previous research [22], however, citrus fruits were never included and assessed for PAT presence. Furthermore, Hussain et al. [27] investigated the natural presence of PAT in orange fruit, juice, pulp, and orange jam samples. A comprehensive survey was conducted in the undertaken study, and the information would be useful for farmers, traders, and local consumers.

### 3.3. Daily Intake Assessment of Patulin

The dietary intake of PAT from citrus fruits in Pakistani Punjab is presented in Table 5. The daily intake of PAT varies in the range of 0.10–1.11 µg/kg bw/day. The highest daily intake of PAT was 1.11 µg/kg bw/day in Multan city from the consumption of highly contaminated mosambi fruits. The levels of dietary intake of PAT from the consumption of citrus fruits in northern Pakistan are presented in Table 6. The levels of daily PAT intake varied from 0.13 to 1.93 µg/kg bw/day. The maximum dietary intake level was 1.93 µg/kg bw/day of sweet orange in Sawt city. Rahimi and Jeiran [36] estimated the daily intake of PAT from fruit juice at 16.4, 45.9, and 74.6 ng/kg bw/day in Iranian adults, children, and babies, respectively, two orders of magnitude lower than the levels estimated in the present research. Relatively low values of PAT daily intake were estimated in a previous study by our group, analyzing the effect of consumption of fruit, juices and smoothies [22]. In contrast, PAT dietary intake from apple juice in Indian consumers was in the range of 0.11–0.24 ug/kg bw/day [35], more in line with the values reported here. In China, Guo et al. [37] estimated dietary intakes among adults, children, and babies of apple juice and found levels of 28.1, 67.5, and 110 ng/kg bw/day, respectively. Piemontese et al. [38], from Italy, reported the daily intake of PAT in different age groups and documented levels ranging from 0.22 to 3.41 ng/kg bw/day, with a mean consumption of 21 g per day in different fruits. The limitation of dietary intake results is that dietary intake was conducted in male participants only, and their education level was not included, which could fluctuate the dietary assessment results.

## 4. Conclusions

The present study demonstrates the extensive PAT contamination of citrus fruits from Punjab and northern Pakistan, with values above the EU limit in about 22% of the samples from both areas and, at the same time, with significant differences between the two areas. A dietary intake of 1.11 and 1.93 µg/kg bw/day of mosambi (Multan) and sweet orange (Swat) was estimated, respectively. The high levels of PAT in citrus fruits could pose a significant health hazard for local consumers. Therefore, the monitoring of fruits and their products at programmed intervals would be highly desirable. Concurrently, farmers, traders, exporters, and consumers should receive adequate health hazard and prevention measures.

## Figures and Tables

**Figure 1 ijerph-18-02270-f001:**
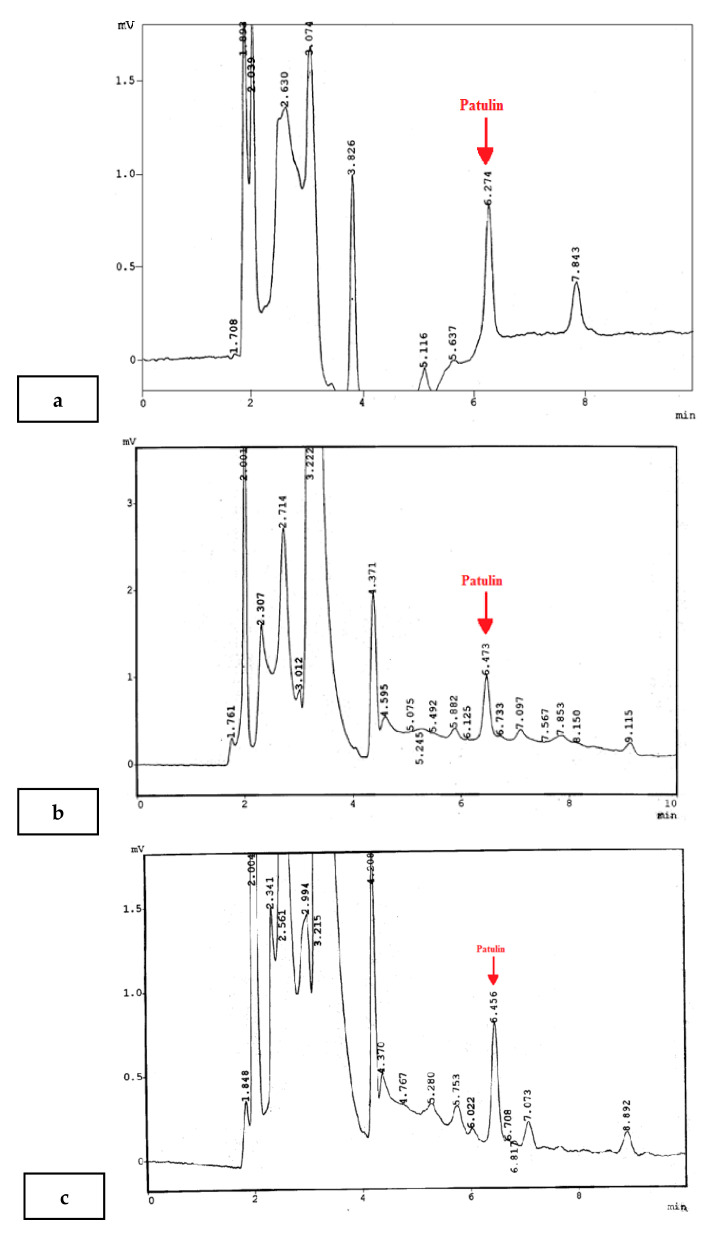
Chromatogram showing the natural occurrence of patulin in orange (**a**), sweet orange (**b**), lime (**c**), and grapefruit (**d**).

**Table 1 ijerph-18-02270-t001:** Recoveries analysis of patulin (PAT) in citrus fruits.

Patulin Fortified Level (µg/kg)	Recovery	RSD	Precision		Retention Time	Coefficient of Determination	LOD	LOQ
	(%)	(%)	Repeatability	Reproducibility	(min)	R^2^	(µg/kg)	(µg/kg)
50	85.5	12.5	14	12	6.437 ± 0.050	0.9947	0.04	0.12
100	89.7	10.6	10	11				
200	91.5	14.5	14	18				

RSD = relative standard deviation, LOD = limit of detection, LOQ = limit of quantification. R^2^ = coefficient of determination; LOQ = LOD × 3; repeatability and reproducibility are given as mean percent RSD (%).

**Table 2 ijerph-18-02270-t002:** Incidence levels of patulin in selected citrus fruit samples from central cities of Punjab, Pakistan.

Citrus Fruits Type	Toba Tek Singh			Sargodha			Multan			Jhang		
	Total/Positive (Positive %)	Mean µg/kg	Range µg/kg	Total/Positive (Positive %)	Mean µg/kg	Range µg/kg	Total/Positive (Positive %)	Mean µg/kg	Range µg/kg	Total/Positive (Positive %)	Mean µg/kg	Range µg/kg
Kinnow	45/22 (49)	89.9 ± 5.36	0.12–130	50/26 (52)	90.9 ± 6.32	0.12–140	40/22 (55)	104.5 ± 4.8	0.12- 127	50/30 (60)	91.7 ± 5.6	0.12- 115
Orange	35/18 (52)	140.4 ± 6.70	0.12–220	40/19 (48)	150.6 ± 7.21	0.12–225	44/25 (54)	142.6 ± 4.36	0.12–245	55/32 (58)	150.3 ± 7.3	0.12–221
Grapefruit	55/22 (40)	150.9 ± 7.6	0.12–190	45/23 (51)	145.7 ± 8.1	0.12–198	50/26 (52)	125.8 ±7.5	0.12–178	35/10 (29)	115.9 ± 4.3	0.12–216
Bitter orange	40/16 (40)	190.6 ± 9.7	0.12–230	35/12 (34)	195.2 ± 10.4	0.12–232	30/15 (50)	198.4 ± 6.1	0.12–210	25/5 (20)	175.2 ± 4.7	0.12–198
Mosambi	45/19 (42)	170.6 ±11.7	0.12–210	40/25 (63)	165.3 ± 10.9	0.12–200	50/28 (56)	194.3 ± 12.5	0.12–295	52/30 (58)	165.6 ± 10.3	0.12–201
Red blood	30/10 (67)	89.9 ± 13.7	0.12–150	50/30 (60)	76.1 ± 12.8	0.12–153	45/22 (49)	75.4 ± 10.6	0.12–135	46/25 (54)	77.4 ± 8.4	0.12–111
Pineapple	20/10 (50)	45.7 ± 15.7	0.12–120	30/15 (50)	50.2 ± 15.2	0.12–121	25/9 (36)	54.1 ± 14.3	0.12–110	20/9 (45)	65.7 ± 11.5	0.12–113
Sweet orange	30/17 (57)	55.7 ± 9.8	0.12–140	40/22 (55)	65.4 ± 10.1	0.12–155	45/26 (58)	45.2 ± 7.5	0.12–1150	25/6 (24)	55.2 ± 8.5	0.04–99
Rough lime	48/27 (56)	78.6 ± 10.7	0.12–150	45/27 (60)	65.3 ± 9.3	0.12–168	30/11 (37)	88.3 ± 9.6	0.12–146	30/12 (40)	97.4 ± 11.3	0.12–123
Sweet lime	50/28 (56)	99.7 ± 15.8	0.12–205	55/33 (60)	104.6 ± 14.9	0.12–210	30/8 (27)	103.1 ± 12.5	0.12–225	22/5 (23)	122.6 ± 8.9	0.12–197
Kagzi lime	43/22 (51)	110.6 ± 11.7	0.12–220	45/32 (71)	122.5 ± 10.4	0.12–234	55/30 (55)	125.4 ± 15.3	0.12–215	53/35 (66)	130.1 ± 14.3	0.12–231
Lemon	40/26 (65)	85.5 ± 16.7	0.12–130	50/30 (60)	92.3 ± 15.9	0.12–145	54/36 (67)	106.7 ± 15.3	0.12–155	50/32 (64)	125.5 ± 9.6	0.12–147
Total	481/247 (51.4)		0.12–230	525/294 (56)		0.12–234	498/329 (66)		0.12–1150	463/231 (50)		0.12–231
Total	1967/1101 (56.0)											

Positive % = positive samples percentage.

**Table 3 ijerph-18-02270-t003:** Incidence levels of patulin in selected citrus fruit samples from northern areas of Pakistan.

Citrus Fruits Type	Mirpur			Peshawar			Swat		
	Total/Positive (Positive %)	Mean µg/kg	Range µg/kg	Total/Positive (Positive %)	Mean µg/kg	Range µg/kg	Total/Positive (Positive %)	Mean µg/kg	Range µg/kg
Kinnow	45/24 (53.3)	75.3 ± 4.1	0.12- 178	52/31 (59.7)	80.2 ± 9.6	0.12- 167	25/10 (40)	80.5 ± 7.3	0.12- 320
Orange	40/16 (40)	99.4 ± 8.5	0.12–156	44/22 (50)	144.2 ± 9.2	0.12–167	25/5 (20)	163.3 ± 10.1	0.12–204
Grapefruit	20/6 (30)	76.2 ± 11.4	0.12–187	25/3 (12)	88.4 ± 6.3	0.12–168	10/1 (10)	109.6 ± 5.8	0.12–121
Bitter orange	10/1 (10)	102.8 ± 9.4	0.12–199	16/1 (6.2)	133.4 ± 10.5	0.12–186	05/1 (20)	156.3 ± 6.3	0.12–209
Mosambi	33/8 (24.2)	111.4 ± 11.7	0.12–150	35/7 (20)	125.8 ± 15.2	0.12–221	40/23 (57.5)	145.2 ± 12.1	0.12–178
Red blood	30/6 (20)	72.1 ± 9.4	0.12–187	40/18 (45)	67.3 ± 3.8	0.12–196	20/6 (30)	56.7 ± 5.2	0.12–120
Pineapple	15/3 (20)	45.2 ± 11.8	0.12–165	15/2 (13.3)	36.4 ± 8.2	0.12–178	20/5 (25)	53.1 ± 13.7	0.12–101
Sweet orange	15/2 (13.3)	56.3± 8.8	0.12–133	14/1 (7.1)	65.2 ± 6.2	0.12–130	20/4 (20)	67.4 ± 12.4	0.12–102
Rough lime	30/10 (3.3)	88.9 ± 13.6	0.12–156	25/5 (20)	95.3 ± 12.7	0.12–168	35/13 (37.1)	86.9 ± 4.5	0.12–132
Sweet lime	20/4 (20)	56.8 ± 10.7	0.12–203	22/8 (36.3)	44.1 ± 5.3	0.12–199	27/10 (37)	35.8 ± 4.2	0.12–210
Kagzi lime	45/25 (55.5)	109.6 ± 8.7	0.12–225	40/21 (52.5)	125.4 ± 13.9	0.12–208	40/20 (50)	156.3 ± 11.4	0.12–234
Lemon	40/21 (52.5)	114.5 ± 15.8	0.12–221	33/14 (42.4)	125.2 ± 7.9	0.12–199	32/15 (47)	111.9 ± 13.4	0.12–178
Total	343/126 (36.7)		0.12–225	361/133 (36.8)		0.12–221	299/113 (37.8)		0.12–320
Total	1003/372 (37.1)								

Positive % = positive samples percentage.

**Table 4 ijerph-18-02270-t004:** The percentage of samples of Patulin exceeds the recommended limits of the European Union.

Citrus Fruits Type	Toba Tek Singh		Sargodha		Multan		Jhang		Mirpur		Peshawar		Swat	
	n ≥ 50 µg/kg	N (%) ≥ 50 µg/kg	n ≥ 50 µg/kg	N (%) ≥ 50 µg/kg	n ≥ 50 µg/kg	N (%) ≥ 50 µg/kg	n ≥ 50 µg/kg	N (%) ≥ 50 µg/kg	n ≥ 50 µg/kg	N (%) ≥ 50 µg/kg	n ≥ 50 µg/kg	N (%) ≥ 50 µg/kg	n ≥ 50 µg/kg	N (%) ≥ 50 µg/kg
Kinnow	12	26.7	14	28.0	10	25.0	18	36.0	13	28.9	14	26.9	5	20.0
Orange	11	31.4	10	25.0	12	27.3	16	29.1	9	22.5	10	22.7	0	0.0
Grapefruit	10	18.2	12	26.7	20	40.0	5	14.3	1	5.0	0	0.0	0	0.0
Bitter orange	9	22.5	8	22.9	10	33.3	0	0.0	0	0.0	0	0.0	0	0.0
Mosambi	12	26.7	15	37.5	16	32.0	14	26.9	2	6.1	0	0.0	8	20.0
Red blood	7	23.3	18	36.0	11	24.4	12	26.1	0	0.0	7	17.5	0	0.0
Pineapple	7	35.0	7	23.3	4	16.0	4	20.0	0	0.0	0	0.0	0	0.0
Sweet orange	11	36.7	10	25.0	6	13.3	2	8.0	0	0.0	0	0.0	0	0.0
Rough lime	13	27.1	13	28.9	5	16.7	4	13.3	2	6.7	0	0.0	7	20.0
Sweet lime	14	28.0	16	29.1	4	13.3	0	0.0	0	0.0	0	0.0	5	18.5
Kagzi lime	12	27.9	18	40.0	13	23.6	14	26.4	6	13.3	10	25.0	13	32.5
Lemon	14	35.0	15	30.0	15	27.8	14	28.0	9	22.5	8	24.2	11	34.4
Total	132	27.4	156	29.7	126	25.3	103	22.2	42	12.2	49	13.6	49	16.4
Total	657 (22.1) ^a^													

n = number of samples; N (percentage of individual samples to the total number of samples). ^a^ = total samples of all regions higher than EU regulations (percentage of total samples of all regions, higher than EU regulations).

**Table 5 ijerph-18-02270-t005:** Dietary intake of patulin in citrus fruit samples from central cities of Punjab, Pakistan.

Citrus Fruits Type		Toba Tek Singh		Sargodha		Multan		Jhang	
	Consumption	Mean	Patulin Intake	Mean	Patulin Intake	Mean	Patulin Intake	Mean	Patulin Intake
	g/day	(µg/g)	µg/kg bw/day	(µg/g)	µg/kg bw/day	(µg/g)	µg/kg bw/day	(µg/g)	µg/kg bw/day
Kinnow	200	79.9	0.23	80.9	0.23	84.5	0.24	81.7	0.23
Orange	250	140.4	0.51	150.6	0.54	142.6	0.51	150.3	0.54
Grapefruit	100	150.9	0.22	145.7	0.21	125.8	0.18	115.9	0.17
Bitter orange	300	190.6	0.82	195.2	0.84	198.4	0.85	175.2	0.75
Mosambi	400	170.6	0.97	165.3	0.94	194.3	1.11	165.6	0.95
Red blood	250	89.9	0.32	76.1	0.27	75.4	0.27	77.4	0.28
Pineapple	150	45.7	0.10	50.2	0.11	54.1	0.12	65.7	0.14
Sweet orange	1000	55.7	0.80	65.4	0.93	45.2	0.65	55.2	0.79
Rough lime	400	78.6	0.45	65.3	0.37	88.3	0.50	97.4	0.56
Sweet lime	200	99.7	0.28	104.6	0.30	103.1	0.29	122.6	0.35
Kagzi lime	100	110.6	0.16	122.5	0.18	125.4	0.18	130.1	0.19
Lemon	100	85.5	0.12	92.3	0.13	106.7	0.15	125.5	0.18

Body average weight = 70 ± 2.

**Table 6 ijerph-18-02270-t006:** Dietary intake of patulin in citrus fruit samples from northern areas of Pakistan.

Citrus Fruits Type		Mirpur		Peshawar		Swat	
	Consumption	Mean	Patulin Intake	Mean	Patulin Intake	Mean	Patulin Intake
	g/day	(µg/g)	µg/kg bw/day	(µg/g)	µg/kg bw/day	(µg/g)	µg/kg bw/day
Kinnow	240	95.3	0.33	101.2	0.35	90.5	0.31
Orange	270	99.4	0.38	144.2	0.56	163.3	0.63
Grapefruit	200	76.2	0.22	88.4	0.25	109.6	0.31
Bitter orange	350	102.8	0.51	133.4	0.67	156.3	0.78
Mosambi	300	111.4	0.48	125.8	0.54	145.2	0.62
Red blood	350	72.1	0.36	67.3	0.34	56.7	0.28
Pineapple	250	45.2	0.16	36.4	0.13	53.1	0.19
Sweet orange	2000	56.3	1.61	65.2	1.86	67.4	1.93
Rough lime	300	88.9	0.38	95.3	0.41	86.9	0.37
Sweet lime	400	56.8	0.32	44.1	0.25	35.8	0.20
Kagzi lime	200	109.6	0.31	125.4	0.36	156.3	0.45
Lemon	300	134.5	0.58	145.2	0.62	131.9	0.57

Body average weight = 70 ± 2.

## Data Availability

The data can be made available when requested.

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
