# Peer review of "Patulin Contamination of Citrus Fruits from Punjab and Northern Pakistan and Estimation of Associated Dietary Intake"

_ijerph, 2021, doi:10.3390/ijerph18052270_

Round 1
Reviewer 1 Report
In the manuscript titled “Variation of patulin levels in citrus fruits from central cities of Punjab and Northern cities of Pakistan, and estimation of dietary intake” Aslam et al. analyzed a high number of samples, monitoring the presence of natural occurrence of patulin, a known mycotoxin. The authors described the presence of high amounts of this mycotoxin in samples from central cities of Punjab and Pakistan's Northern cities. The manuscript appears interesting and give important information about the current situation in these cities, promoting new land managements.
However, I have some concerns about the determination of patulin in samples, because authors describing the HPLC methodology did not report the use of an autosampler, this mean great variations in the retention times used for the determination of components. If the retention time of the target component differs between the standard and the sample, frequently in complex matrices, the target component is identified: a) by adding the standard sample to the unknown sample; or b) by adding/using a different internal standard as reference; or c) analyzing the HPLC-collected components, i.e. by IR or mass spectroscopy.
I.e. in Spadaro et al., (using autosampler) the authors used the presence of 5-hydroxymethylfurfural, a common compound of apple juice, eluting just prior to patulin, as a standard reference for the patulin retention time. In Piemontese et al., authors (using autosampler) compared the retention time of patulin peaks in contaminated samples with respect to that of authentic standard of patulin, but an additional confirmation was also performed in selected positive samples that were reanalysed by HPLC after spiking the extracts with an equivalent amount of patulin by checking the symmetry of the peak eluting at retention time of patulin.
I’m sure that authors have verified their results, so I ask them to report their standard controls, it is better as supplementary materials, in order to support their data. This is very important, because they found very high amount of patulin in their samples, having a high impact on the people health management and also on the production chain and trade of these products. Thus, I suggest to publish after major revision.
Minor comments
Page 3 row 125: I suggest to change in: “PAT's standard curve was made by a dilution with methanol of PAT in the range of concentration from 0.4 to 300 μ g/L, and stored in sealed vials at -20 °C, till further analysis.”
Page 3 row 132: authors should give more information about the sample characteristic, in particular, is it just the peel or also part of edible fruit?
Page 3 rows 133-134: I suggest to change in: “... ethyl acetate was added and PAT extracted ...”
Page 3 row 138: I suggest to change in: “... pH was maintained at the value of 4.0 with ...”
Page 3 row 139: authors should check °C, because sometime is underlined or superscript zero.
Page 3 row 140: about the sentence “... and purified with a 0.22 mm syringe filter ...” more precisely, it is not a purification, but particle removal and/or solution clarification, so I suggest to modify.
Pages 3-4 rows 143-148: in the paragraph 2.4 HPLC conditions, both column temperature and detection wavelength are missed.
Page 4 rows182-184: it is not clear if the values of 0.04 (LOD) and 0.12 μg/kg (LOQ) of patulin are from standard curve or sample matrix?
Table 1: some data at 100 and 200 μg/Kg are missed in the table.
Tables 2 and 3: in these tables authors reported as lower range 0.12 μg/kg, this mean that is patulin detected in all samples? Or was in some sample undetectable? Anyway, represents this the lower value detected in all samples?
Row 243: are authors sure about the value of 113342 μg/kg? I did not find this value in the reference 28. I suggest to verify.
Row 292-293: authors report a consumption of 21 g per day, is it referred to patulin? Are they sure about? Or refer they on the mean apple juice intakes of 200 and 216 g day, as reported in the reference 38?
Author Response
The file is attached

Reviewer 2 Report
Aslam Kinza et al. (2021). Variation of patulin levels in citrus fruits from central cities of Punjab and Northern cities of Pakistan, and estimation of
Aslam Kinza et al. (2021). Variation of patulin levels in citrus fruits from central cities of Punjab and Northern cities of Pakistan, and estimation of dietary intake
Suggested title:
Patulin contamination of citrus fruits from Punjab and Northern Pakistan, and estimation of associated dietary intake
The paper is fully coherent with the scope of the Journal and provides useful novel information on the occurrence of patulin (PAT), a widespread mycotoxin, in numerous citrus varieties cultivated in Pakistani Punjab and Northern Pakistan. In addition, the paper provides an estimation of PAT daily intake by local population, as vehiculated by the consumption of citrus fruits.
The technique employed is sound and the statistical analysis of data is adequate. There is only a point in the choice of surveyed individuals that requires attention (see below)
A major problem with this paper is in the presentation form. Starting from the title, the manuscripts is verbose, frequently unclear, and repetitive. I am attaching the pdf with numerous suggestions included as notes. I invite the authors to read the notes and revise the manuscript accordingly.
I also have a few, more specific observations:
Lines 85-86: re-write the sentence: in the present form it is unintelligible.
Line 95: give full name of JECFA
Line 158: I found it strange that the weight of individuals surveyed for eating habits varied in such a narrow range. It seems that they belong to an extremely homogeneous, namely not fully representative group. All males? Why? This is a crucial point that defines the soundness of the work and therefore needs being addressed carefully.
Lines 218-220: the sentence is obscure to me. Please re-write it in a better form.
Starting from the title, the authors refer to cities as the places of origin of the citrus samples. I would avoid this, referring instead to geographic areas (Pakistani Punjab and Northern Pakistan), and mentioning cities only to circumscribe specific subareas.
Last, the authors frequently use the expression “selected fruits”. This appears to suggest that they chose the fruits for analysis, possibly selecting damaged ones or viceversa, which obviously they did not. What they mean is "selected citrus cultivars or varieties". I chose to delete “selected” throughout. With my observation in mind, the authors might choose another solution.

Author Response
The file is attached

Reviewer 3 Report
Patulin is a mycotoxin produced by various fungal species of the genus Penicillium and Aspergillus which can be found in damaged fruit types and their derivatives, such as juices and drinks.
The aim of the work of Aslam and coworkers was to determine the possible occurrence of patulin in 12 types of citrus fruits from some cities of Punjab and Pakistan.
General comments
The topic is not particularly original, but the authors performed extensive screening on a total of 2970 samples of citrus fruits. Furthermore, the work is well structured and the experimental part is conducted correctly, allowing to draw the right conclusions. On the other hand, the data from this research could suggest regular monitoring of fruit and fruit products in those countries where there is still no legal regulation on potential toxic contaminants.
However, the work requires some modifications.
Specific comments
Line 95: Enter the full meaning of the abbreviation (JECFA), that is: The Joint FAO/WHO Expert Committee on Food Additives.
Line 122: Check the concentration of the patulin standard: it should be 100 microg/ml.
Line 125: It is known that patulin is a very stable molecule in low pH and resistant to thermal denaturation, in fact, its removal in food through pasteurization is very difficult. Why was patulin diluted in methanol, if it is more stable in an acidic medium (as also reported by the same authors, line 85)?
Line 185: Replace the sentence "The accuracies of both these quantities mainly depend on the sensitivity of the instrument" with "The accuracies of both these quantities mainly depend on the sensitivity of the method".
Author Response
The file is attached

Reviewer 4 Report
1) The list of references contains 38 items, while the text includes item 45 line 59. 2) In Tables 2 and 3 and in the reference list - spacing correction, punctuation marks, e.g.3) It should be remembered that each survey method assessing the consumption of certain products has its limitations.
This is worth mentioning.
4) The manuscript does not contain precise information about the research group: age, sex or education.
It is known that these factors affect the diet, including the quantity and quality of consumed products.
Author Response
The file is attached
